# R-PreNet: Deraining Network Based on Image Background Prior

**Congyu Jiao, Fanjie Meng \*, Tingxuan Li and Ying Cao**

Institute of Intelligent Control and Image Engineering, Xidian University, Xi'an 710071, China; 21131213388@stu.xidian.edu.cn (C.J.); 21131213389@stu.xidian.edu.cn (T.L.); 22131214134@stu.xidian.edu.cn (Y.C.)
\* Correspondence: fjmeng@xidian.edu.cn

**Abstract:** Single image deraining (SID) has shown its importance in many advanced computer vision tasks. Although many CNN-based image deraining methods have been proposed, how to effectively remove raindrops while maintaining background structure remains a challenge that needs to be overcome. Most of the deraining work focuses on removing rain streaks, but in heavy rain images, the dense accumulation of rainwater or the rain curtain effect significantly interferes with the effective removal of rain streaks, and often introduces some artifacts that make the scene more blurry. In this paper, a novel network architecture, R-PReNet, is introduced for single image denoising with an emphasis on preserving the background structure. The framework effectively exploits the cyclic recursive structure inherent in PReNet. Additionally, the residual channel prior (RCP) and feature fusion modules have been incorporated, enhancing denoising performance by emphasizing background feature information. Compared with the previous methods, this approach offers notable improvement in rainstorm images by reducing artifacts and restoring visual details.

**Keywords:** single image deraining; residual channel prior; interactive fusion

## 1. Introduction

Rainfall is a prevalent meteorological phenomenon [1] that adversely affects the visual quality of images and hampers the performance of subsequent image processing tasks such as object recognition [2], object detection [3], autonomous driving, and video surveillance [4–6]. Consequently, the removal of rain streaks from rainy images has emerged as a significant and meaningful research topic, gaining attention in recent years. Single-image deraining refers to the restoration of a clean, rain-free image scene from a rainy single image. However, given the intricate amalgamation of background information and raindrop details, simultaneously eliminating the raindrops and preserving the background remains a challenging issue. We found in an experiment that the PReNet deraining network model [7] can reconstruct a relatively clear rain-free image, but in the test of a rainstorm dataset, the background structure of the reconstructed image corresponding to the rainstorm image has also been damaged to some extent, that is, the introduction of artifacts, and the destruction of the image background can sometimes lead to serious problems, such as blurry or missing traffic signs, which may result in serious accidents in autonomous driving. In order to address this problem, this paper introduces an additional image background prior to protect the background structure, so that a clearer and correct reconstruction of rainless images can be obtained in the case of processing rainstorm images, as shown in Figure 1.

In this article, we explore the effective reconstruction problem of complex combinations of background and raindrops, and propose a new algorithm called R-PReNet that can effectively remove raindrops and protect background information. This algorithm fully utilizes the cyclic recursive structure of PReNet and its capability to remove rain streaks. On this basis, this article introduces residual channel prior (RCP) [8–11] in the model to achieve background structure protection. In addition, this article also proposes the use of the 'Squeeze Excitation' residual module (SE ResBlock) [12] to extract deep features of

RCP, and the interactive fusion feature module (IFM) [11] to fully utilize RCP information, achieving high-quality rainless image reconstruction.

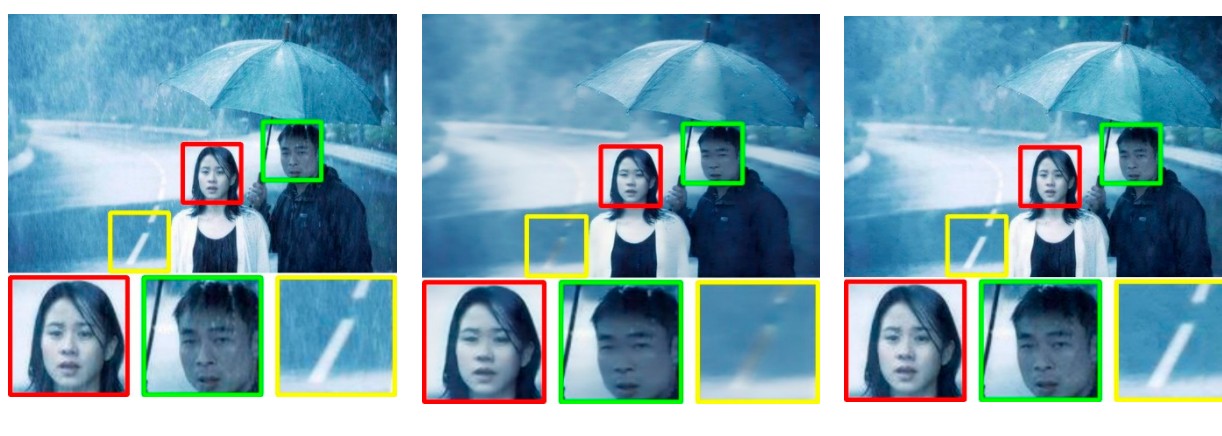

(**a**) Real rain image                        (**b**) PReNet deraining results                        (**c**) R-PReNet deraining results

**Figure 1.** Image deraining in the real world. PReNet [7] and R-PReNet were trained on RainTrainH. (**a**) is a real rain image, (**b**) is the result image after using PReNet to remove rain, and (**c**) is the result image after using this algorithm to remove rain. This images show that R-PReNet can effectively remove rain streaks while retaining better background textures and maintaining the basic tone of the original image.

Our contributions are summarized as follows:

This article replicates and tests the PReNet deraining network on three popular image deraining datasets (Rain100H [13], Rain100L [13], Rain14000 [14]) and real rainy image datasets (Practical_by_Yang [13]), and studies the results of deraining.

This article explores the effectiveness of residual channel prior (RCP) for background protection and proposes an image deraining network structure based on RCP. Numerous experiments have shown that our method outperforms the original method on commonly used rainfall datasets, restoring visually clean images and good detail.

An RCP extraction module and an interactive fusion module (IFM) are introduced, designated for RCP extraction and guidance, respectively. These aim to attain deep features of the RCP and guide the network to recover more background details.

The remainder of this paper is structured as follows. Section 2 briefly reviews relevant studies on image denoising methods. Section 3 presents the comprehensive R-PReNet denoising network based on image background prior and delves into the RCP residual channel prior and IFM fusion techniques. Experimental results and comparisons are detailed in Section 4. The conclusion is given in Section 5.

## 2. Related Works

The objective of the single-image deraining task is to recover a rain-free image from its rain-corrupted counterpart. The video-based deraining task can use video to obtain consecutive multiple frames of images, and use the temporal nature of the continuous images to obtain the position information of rain streaks and the background information of rain streaks occlusion positions, thus achieving the video-based deraining task. However, the task of removing rain from a single image is challenging due to the absence of information regarding the positions of rain streaks and the occluded background, complicating the reconstruction of a rain-free image.

Existing approaches proposed for this task can be primarily categorized into two classes: model-driven methods and data-driven methods.

### 2.1. Model-Driven Methods

Generally speaking, early filter-based methods and traditional prior based methods belong to the model-driven type. We will introduce representative works of these two aspects as follows.

An image can be decomposed into low-frequency and high-frequency parts, with details and noise information predominantly located in the high-frequency part. Consequently, it is evident that raindrops in a rainy image are mainly distributed in its high-frequency portion. Thus, in the initial stages of rain removal from single images, guided filters [15] have been introduced as a universal tool for image prior representation, which decomposes the rainy image into its low-frequency part (LFP) and high-frequency part (HFP). Subsequently, Xu et al. [16], Zheng et al. [17], Ding et al. [18], and Kim et al. [19] employed the characteristics of rain streaks and various guided filtering methods for single rain image deraining, achieving preliminary success. However, there are still issues such as leaving obvious rain streaks and missing background details, so there is room for further performance improvement in this method.

Broadly speaking, rain-affected images are considered to be composed of a background layer and a rain layer:

$$O = B + S \tag{1}$$

where B denotes the background layer, which represents the target image to be obtained; S symbolizes the rain streak layer; and O represents the input image with rain traces. Thus, the problem of rain removal can be formulated as an image decomposition issue based on dictionary learning and sparse representation. Therefore, scholars no longer only rely on different guided filtering methods to remove rain from a single rain image, but have begun to study the physical properties of rain streaks themselves (such as sparsity and self-similarity), and introduced them into the deraining model as prior information, thus realizing the reconstruction of rainless images.

Kang et al. [20,21] initially employed a bilateral filter to decompose the image into high-frequency and low-frequency components. Subsequently, the high-frequency component was further decomposed into "rainy components" and "non-rainy components" using dictionary learning and sparse coding. The rainy component was then removed from the image, preserving the majority of the original image details. This algorithm emphasizes training within the high-frequency layer rather than within the image domain, offering advantages in reduced computational resources and undisturbed low-frequency layer processing. However, this method is time-consuming, and, due to its heavy reliance on the bilateral filter preprocessing, the background is typically blurred, suggesting room for further performance optimization.

In order to further obtain a clear background layer, a comprehensive exploration of the intrinsic properties of both the background and raindrops was conducted, and these properties were regularized to constrain the solution space. The classic methods include (1) The emergence of low rank due to the non-local similarity of raindrops [22]; (2) the Gaussian mixture model (GMM) employed for calculating the rain streak distribution across various scales and orientations [23,24]; and (3) a sparse representation model based on some learning rain atoms [25]. Although these methods have modeled and referenced both rain streaks and background layers, they can only handle light rain streaks, making it difficult to handle heavy or sudden rain streaks, and there is still a problem of time-consuming processing.

Research into these model-driven methods revealed that although incorporating physical prior information about rain patterns and background layers can help achieve rain image reconstruction, this prior information is usually subjective and incomplete, making it difficult to fully transfer existing prior knowledge in real rain images [13,24,26]. In particular, the rain images obtained from real scenes are often complex and variable, so the performance of directly establishing models for removing rain is always unsatisfactory. Therefore, data-driven deep learning deraining algorithms have become the latest trend in deraining tasks.

### 2.2. Data-Driven Methods

With the advancement of deep learning theories and techniques, data-driven single image deraining approaches are becoming increasingly prevalent. These methods automatically extract features from the dataset through network structures, thereby achieving mapping from rain images to deraining images.

Since 2013, Eigen et al. [27] trained a special CNN by minimizing the mean square deviation between predicted rain and no rain image blocks, and for the first time, used deep learning methods to remove raindrops attached to images. To this day, numerous CNN-based deep networks for image deraining have been proposed. [13,14,28]. Usually, in these deep neural networks, constraints related to rainfall, such as rainfall masks and background features, are added to the network to learn features more comprehensively. Later, some methods utilized cyclic networks and residual networks [7] to gradually remove raindrops, which streamlined the network structure and reduced network parameters.

However, due to the challenges of obtaining paired real rainy and rain-free images, there exists a disparity between synthesized rainy images and actual rainy photographs. Previous deraining algorithms may result in certain performance deviations when directly applied to real rain images. To address the above issues, experts have considered introducing unsupervised and semi-supervised methods in image deraining networks.

Semi-supervised learning leverages both unlabeled and labeled data for training. For instance, Wei et al. [24] introduced SIRR, a network that simulates genuine rainfall residuals through the likelihood term applied to the Gaussian mixture model, minimizing the Kullback–Leibler divergence between synthesized and real rain distributions. Yasala et al. [29] proposed Syn2real (GP), which uses Gaussian process to model latent features of rainy images and generates pseudo labels for unlabeled data. Huang et al. [30] proposed MOSS, which uses a memory oriented decoder encoder network to comprehend rain patterns and recover rain-free background images. By jointly mining rain streak features from both real and synthesized datasets through various approaches, these methods have enhanced the generalization capability of the deraining algorithms.

Unsupervised learning means that it does not rely on the marked data and directly models the input data. The unsupervised algorithm in the deraining algorithm is implemented through the introduction of generative adversarial networks (GAN). Zhu et al. [31] proposed an unsupervised end-to-end adversarial deraining network termed RainRemoval GAN (RR-GAN), which is capable of generating genuine rain-free images solely using unpaired images. The network is chiefly comprised of a multi-scale attention memory generator and a multi-scale attention discriminator, with its architecture still bearing resemblance to supervised GAN methodologies. Jin [32] proposed another unsupervised generative adversarial network (UD-GAN), which introduced self-supervision constraints into the internal statistical information of unpaired rain and clean images. It uses two mutually cooperative modules, namely the background guidance module (BGM) and the rain guidance module (RGM). The RGM is specifically designed to differentiate between genuine rain-free images and the fake rain-free images generated based on the BGM. The BGM ensures background consistency between the rain-streaked input and the down-sampled output by leveraging a hierarchical Gaussian blur gradient error.

Afterwards, there were also unsupervised algorithms such as DerainCycleGAN [33] which solved the problems of difficulty in obtaining paired real rain images and rainless images, as well as poor generalization ability of algorithms based on synthetic images. However, there are still problems such as overly complex networks, time-consuming training, and insufficient rain pattern removal.

While these data-driven approaches can effectively remove certain rain streaks, they fall short in eliminating all rain streaks under complex scenarios, such as images under heavy rain conditions. Moreover, they often struggle to fully preserve the structural information of the image and may even introduce new artifacts during reconstruction. Therefore, a method that is simple and efficient, removes a large number of raindrops, protects object structures, and improves generalization ability is crucial.

## 3. Proposed Work

In this section, the overall network architecture of the proposed algorithm is presented. The implementation details of the introduced residual channel prior (RCP) are first described. Subsequently, the structure of the progressive recursive network (PReNet), serving as the backbone network, is showcased. Finally, a method for fusing high-dimensional features of the RCP is proposed.

### 3.1. Residue-Progressive Recurrent Network

As shown in Figure 2, R-PReNet consists of two main parts: (i) the RCP feature extraction and fusion module, and (ii) the progressive recurrent network. Features from rainy images are first extracted and then merged with the RCP characteristics. Subsequently, the combined features are concatenated with the image attributes. The components of this approach will be detailed in the following sections.

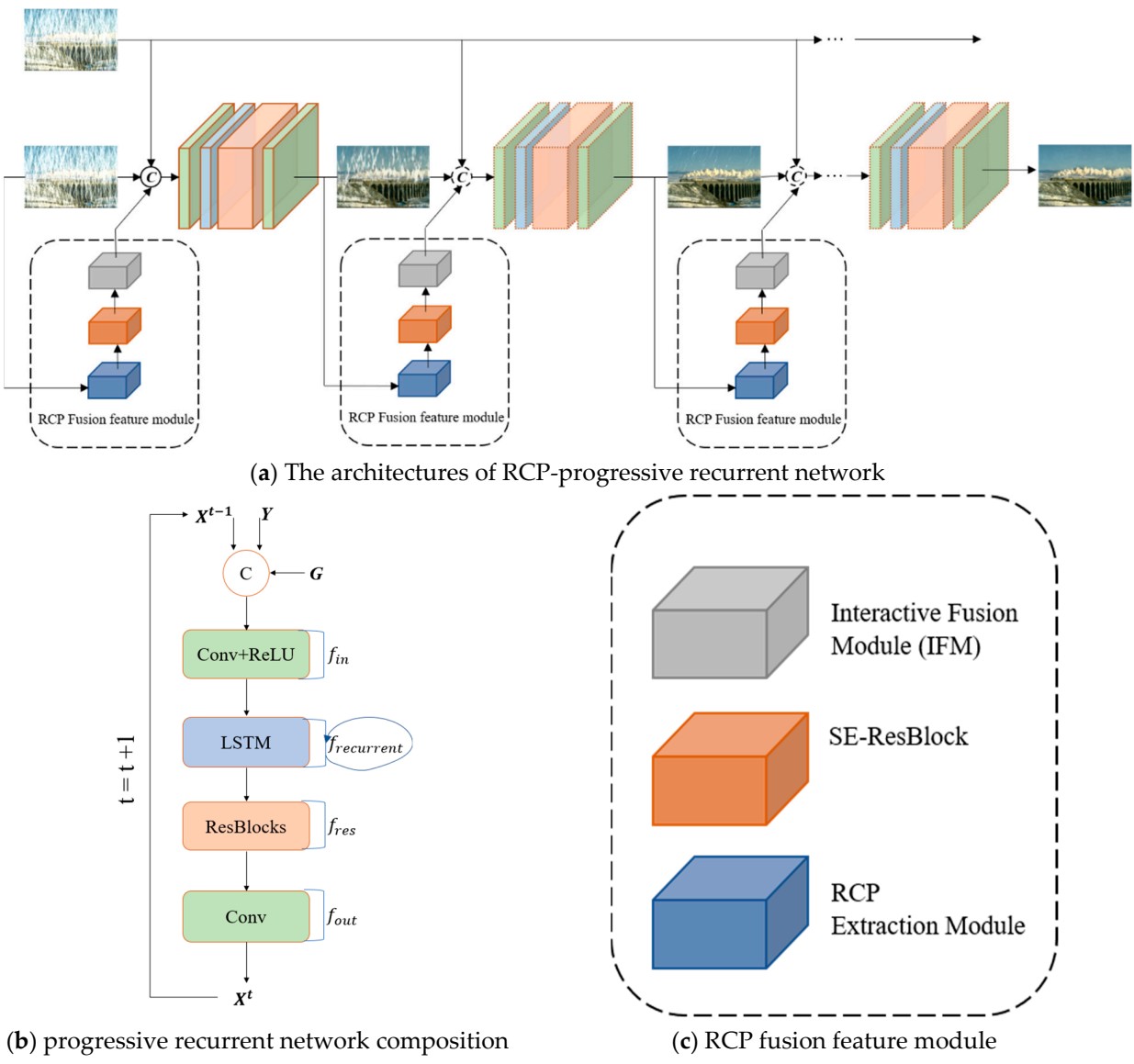

(**a**) The architectures of RCP-progressive recurrent network

(**b**) progressive recurrent network composition  (**c**) RCP fusion feature module

**Figure 2.** The overall structure of residue-progressive recurrent network (R-PreNet), where (**a**) shows the overall network framework of R-PreNet; (**b**) shows progressive recurrent network composition in R-PreNet, where $f_{in}$ is a convolutional layer with ReLU, $f_{res}$ is a recursive ResBlocks, $f_{out}$ is a convolutional layer, $f_{recurrent}$ is a convolutional LSTM, and © is a connectivity layer; (**c**) is the RCP fusion feature module.

### 3.2. Residue Channel Prior (RCP)

The appearance of rain streaks is commonly modeled as a linear combination of the background and rain streak layers [14,20,22,34]. Based on this model, Li et al. [8] demonstrated that subtracting the minimum color channel from the maximum color channel produces a rain-free image. Rain streaks are colorless (white or grey) and appear at the same location in different RGB color channels. As such, subtracting the minimum color channel from the maximum one nullifies the presence of rain streaks, as in Figure 3.

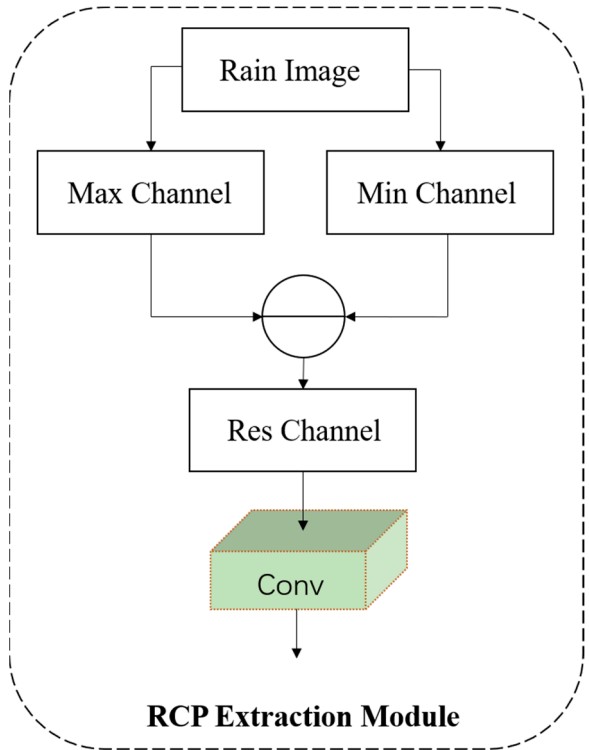

**Figure 3.** RCP extraction module.

The colored-image intensity of a rainy image is defined [8] as:

$$\widetilde{I}(x) = \tau\, \rho_{rs}(x) L\sigma + (T - \tau) B\pi \tag{2}$$

where $\mathbf{L} = \left(L_r, L_g, L_b\right)^T$ is the color vector of luminance and $\mathbf{B} = \left(B_r, B_g, B_b\right)^T$ is the color vector of background reflection.

$$L = L_r + L_g + L_b, \quad B = B_r + B_g + B_b \tag{3}$$

In the model (Equation (2)), the first term represents the rain streak component, while the second term denotes the background component. $\sigma = \mathbf{L}/\,L$ and $\pi = \mathbf{B}/\,B$ define the chromaticities of $\mathbf{L}$ and $\mathbf{B}$. $T$ represents the exposure time, while $\tau$ denotes the time taken by a raindrop to pass through pixel $x$. $\rho_{rs}$ consists of the refraction coefficients of the raindrop, the specular reflection coefficients, and the internal reflection coefficients. The assumption is made that $\rho_{rs}$ is wavelength-independent, implying that raindrops are colorless.

As a consequence, it becomes necessary to cancel the light chromaticity $\sigma$ in the rain-streak term in Equation (2) to generate a residual channel without rain streaks. To achieve this, any existing color constancy algorithm [35] is employed to estimate $\sigma$, and then apply the following normalization step to the input image:

$$I(x) = \frac{\widetilde{I}(x)}{\sigma} = I_{rs}(x)\mathbf{i} + \mathbf{I}_{bg}(x) \tag{4}$$

where $\mathbf{i} = (1, 1, 1)^T$, $I_{rs} = \tau \rho_{rs} L$, $\mathbf{I}_{bg} = (T - \tau)\mathbf{B}/\sigma$.

Vector division is done element-wise. It should be noted that upon normalizing the image, not only is the luminance of light eliminated, but the color effects of spectral sensitivity are also removed. Hence, according to the previous equation and a rainy image **I**, the residual channel is defined as:

$$I_{res}(x) = I^M(x) - I^m(x) \tag{5}$$

where:

$$I^M(x) = \max\{I_r(x), I_g(x), I_b(x)\} \tag{6}$$

$$I^m(x) = \min\{I_r(x), I_g(x), I_b(x)\} \tag{7}$$

$I_{res}$ is the residual channel of the image **I**, which has no rain streaks.

### 3.3. RCP High-Dimensional Feature Extraction

Although the operation of subtracting a color channel from another in the image space is beneficial and the structural information of the RCP is clearer than the rainy image, it can be destructive to the background image because of information loss. Therefore, the operations utilizing the structural information of RCP are shifted to the feature domain. An RCP feature extraction module is introduced to extract the high-dimensional features of the RCP.

Based on the squeeze-and-excitation (SE) block proposed by Hu et al. [36], which focuses on channel relationships to construct informative features, this residual block adaptively recalibrates the channel feature responses by explicitly modeling the interdependencies between channels. Given that the RCP module interacts through color channels, the SE ResBlock structure, as illustrated in Figure 4, is employed to extract the high-dimensional features Fp of the RCP, aiming to reduce noise in the initial features and enrich the semantic information of the features.

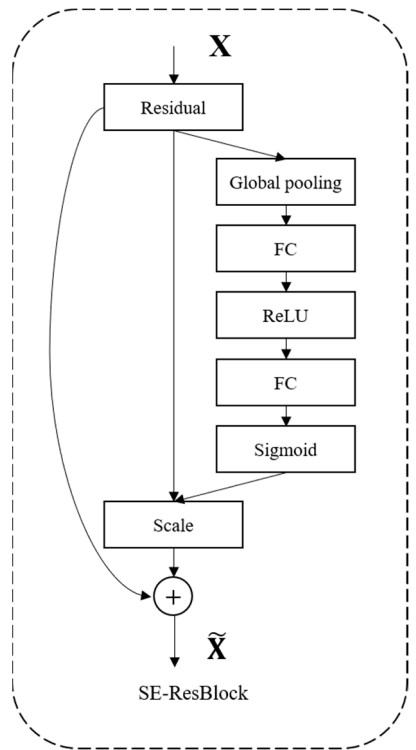

**Figure 4.** SE-ResBlock module.

### 3.4. Interactive Fusion Features

While high-dimensional features of the RCP have been extracted, effectively leveraging these RCP features to guide the model remains a challenging task.

A simple solution is directly concatenating RCP features with image features, but this is ineffective for guiding model deraining and may cause feature interference. To address this problem, an interactive fusion module (IFM) [37] is introduced, consisting of two branches (rainy image features and prior features) to progressively combine features. As shown in Figure 5, two $3 \times 3$ kernel-sized convolutions are performed to map the rainy image features $F_o$ and RCP features $F_p$ to $\hat{F}_o$ and $\hat{F}_p$.

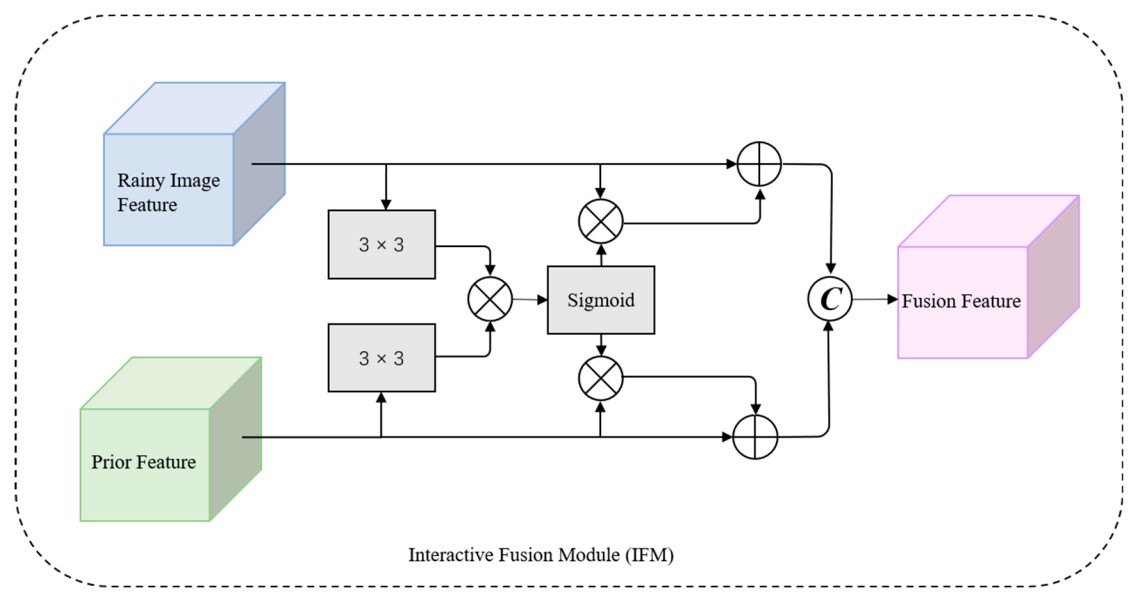

**Figure 5.** Interactive fusion feature module.

Next, the similarity map S between $\hat{F}_o$ and $\hat{F}_p$ is computed using element multiplication:

$$S = \text{Sigmoid}\left(\hat{F}_o \otimes \hat{F}_p\right) \tag{8}$$

The similarity map S is utilized to enhance the background information of rainy images compromised by rain streaks. Furthermore, given that the background of RCP resembles that of the rainy image, the similarity map S also emphasizes feature information in the prior, further bolstering its structural integrity.

### 3.5. Progressive Recurrent Network

The progressive recurrent network consists of the following four parts: (i) a convolutional layer $f_{in}$ receives network inputs, (ii) a recurrent layer $f_{recurrent}$ propagates cross-stage feature dependencies, (iii) several residual blocks $f_{res}$ extract the deep representation, and (ii) a convolutional layer $f_{out}$ outputs deconvolutional results. Where $f_{in}$ takes as input the current estimation $x^{t-1}$, the rainy image y, and the concatenation of the background fusion prior features G. A convolutional long short-term memory (LSTM) is employed for the recurrent layers, given its empirical advantage in image deraining, through which cross-stage feature dependencies can be propagated to facilitate rain streaks removal:

$$x^{t-0.5} = f_{in}\left(x^{t-1}, y, G\right) \tag{9}$$

$$s^t = f_{recurrent}\left(s^{t-1}, x^{t-0.5}\right) \tag{10}$$

$$x^t = f_{out}\left(f_{res}\left(s^t\right)\right) \tag{11}$$

where $f_{in}$, $f_{res}$, and $f_{out}$ are stage-invariant, the network parameters are reused across different stages. The recurrent layer $f_{recurrent}$ takes $x^{t-0.5}$ and the recurrent state $s^{t-1}$ as inputs to stage $t-1$. By unfolding PreNet [7] with T recurrent stages, the deep representation of rain streak removal is favored by recurrent state propagation. The deraining results from the intermediate stages of the network structure indicate that the accumulation of storm streaks can be gradually eliminated.

*3.6. Loss Function*

Given a clean single channel image $I$ and a noisy image $K$ of size $m \times n$, the mean square error (MSE) is defined as:

$$\text{MSE} = \frac{1}{mn} \sum_{i=0}^{m-1} \sum_{j=0}^{n-1} \left[I(i,\ j) - K(i,\ j)\right]^2 \tag{12}$$

On this basis, PSNR (dB) is defined as:

$$\text{PSNR} = 10\log_{10}\left(\frac{MAX_I^2}{\text{MSE}}\right) \tag{13}$$

where $MAX_I^2$ is the maximum possible pixel value of the image. If each pixel is represented by 8 bits of binary, then it is 255. In general, if the pixel value is represented by $B$-bit binary, $MAX_I^2 = 2^B - 1$.

If it is a color image, there are usually three ways to calculate it:

1. Calculate the PSNR of the RGB image's three channels separately and then take the average value.

2. Calculate the MSE of the RGB image's three channels, then divide by 3.

3. Convert the image to YCbCr format, and then only calculate the PSNR of the Y component, which is the brightness component.

Among them, the second and third methods are more common. This algorithm uses the second method.

The peak signal-to-noise ratio (PSNR) is an objective measure of image distortion or noise level. The larger the PSNR value between two images, the more similar it is. The general benchmark is 30 dB, and the deterioration of images below 30 dB is more obvious.

SSIM also describes the similarity of two images, and the formula is measured based on three comparisons between samples x and y: luminance, contrast and structure.

$$l(x,y) = \frac{2\mu_x\mu_y + c_1}{\mu_x^2 + \mu_y^2 + c_1} \tag{14}$$

$$c(x,y) = \frac{2\sigma_x\sigma_y + c_2}{\sigma_x^2 + \sigma_y^2 + c_2} \tag{15}$$

$$s(x,y) = \frac{\sigma_{xy} + c_3}{\sigma_x\sigma_y + c_3} \tag{16}$$

Generally, $c_3 = \frac{c_2}{2}$. where, $\mu_x$ is the mean value of $x$ and $\mu_y$ is the mean value of $y$. $\sigma_x^2$ is the variance of $x$ and $\sigma_y^2$ is the variance of $y$; $\sigma_{xy}^2$ is the variance of $xy$.

$c_1 = (k_1 L)^2$ and $c_2 = (k_2 L)^2$ are two constants to avoid division by zero, and $L$ is the range of pixel values.

$k_1 = 0.01$ and $k_2 = 0.03$ are the default values.

Then:

$$\text{SSIM(x, y)} = \left[ l(x,y)^\alpha c(x,y)^\beta s(x,y)^\gamma \right] \tag{17}$$

During each calculation, an N × M window is taken from the image, and then the window is constantly sliding for calculation. Finally, the average value is taken as the global SSIM.

SSIM specifies the MSSIM of the returned image. This is also a floating-point number between zero and one (the higher the better).

A negative SSIM loss [38] is adopted as the objective function. For a model with **T** stages, there are **T** outputs, $x^1$, $x^2$, ..., $x^{\mathbf{T}}$, with supervision applied only to the final output $x^{\mathbf{T}}$. The negative SSIM loss is:

$$L = -\text{SSIM}\left( x^{\mathbf{T}}, \, x^{gt} \right) \tag{18}$$

where $x^{gt}$ is the corresponding ground-truth clean image.

## 4. Experiments

The model was trained on Ubuntu OS, NVIDIA GeForce GTX 3080Ti GPU using Pytorch framework in Python environment with 12GB of RAM. To validate the effectiveness of the model, evaluations were conducted on three popular image-deraining synthetic datasets (Rain100H, Rain100L, Rain14000) and a real rainy images dataset (Practical_by_Yang) to evaluate our approach:

Combined with the visual effect in the Figure 6 and recognition effect in the Figure 7 of the real rain image, it can be seen that the R-PReNet algorithm has a significant background protection effect. Because the results of task-type evaluation of multi-purpose image deraining (MPID) [39] algorithm on a real dataset show that in most cases, the processing of the rain removal algorithm reduces the recognition accuracy. This paper points out that the rain removal algorithm is not optimized to improve the recognition accuracy in the training process, but some important real semantic information is lost in the rain removal process, which reduces the recognition accuracy. Therefore, the background protection module is added in this algorithm to improve the recognition accuracy.

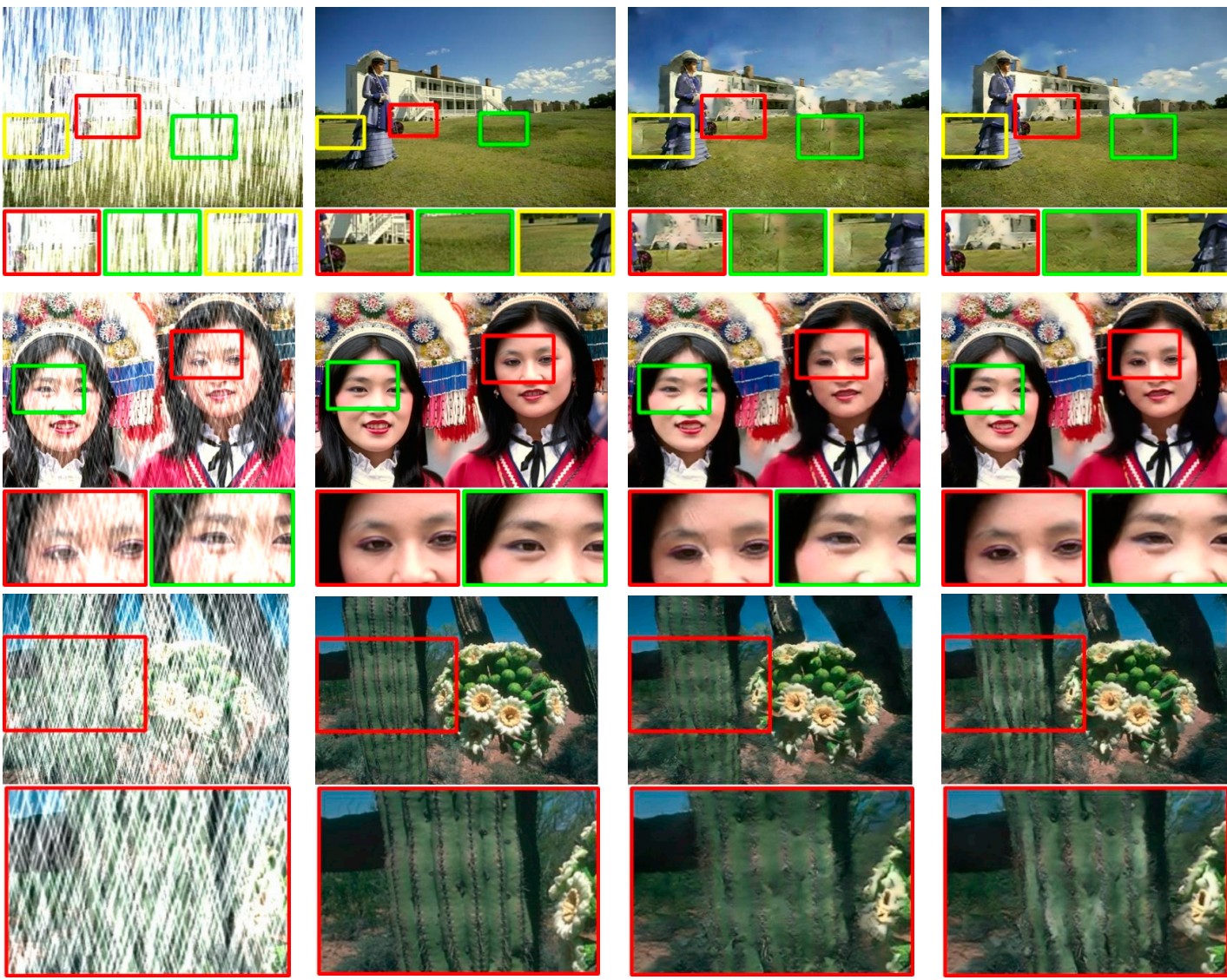

**Figure 6.** *Cont.*

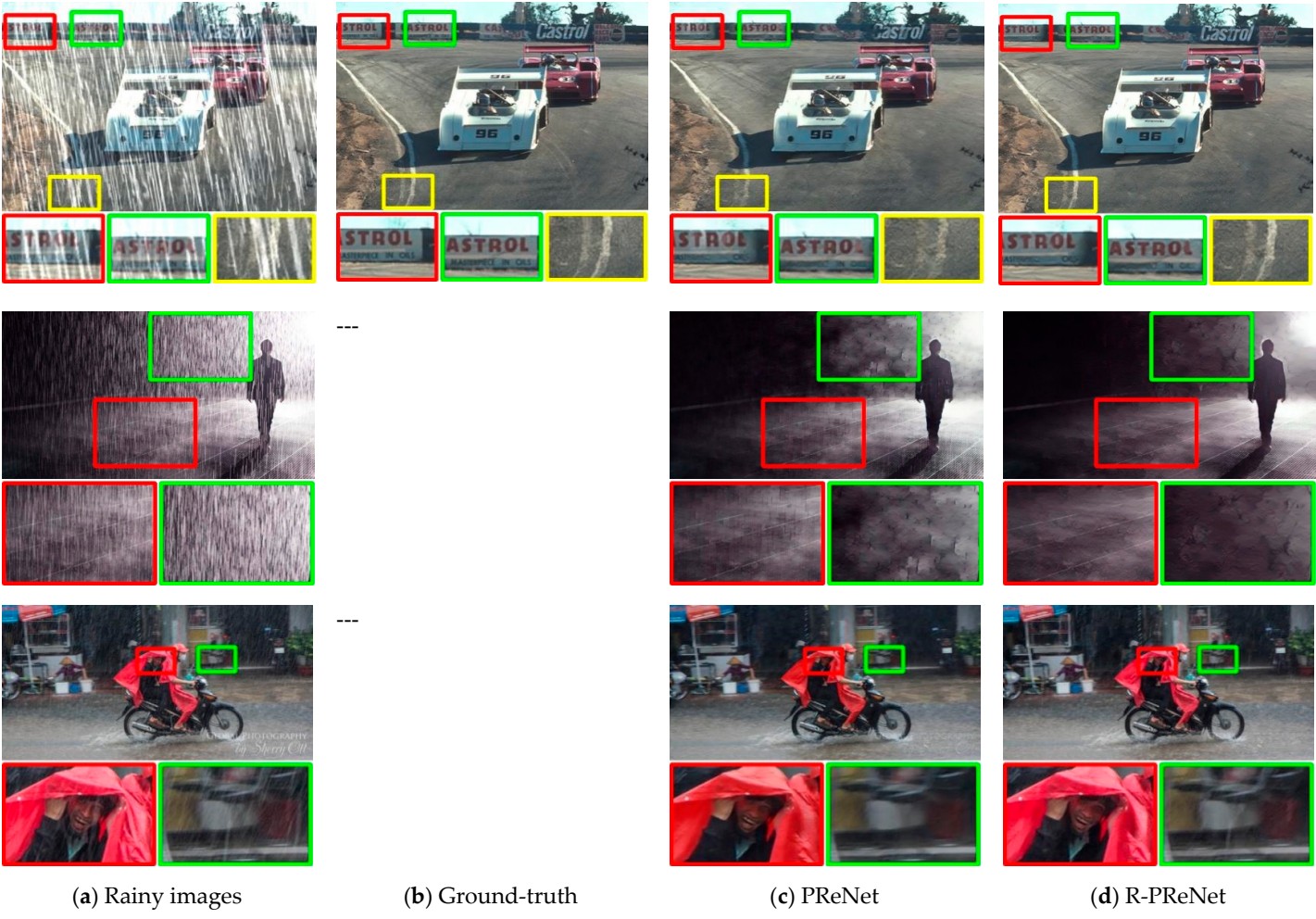

(**a**) Rainy images     (**b**) Ground-truth     (**c**) PReNet     (**d**) R-PReNet

**Figure 6.** Image deraining results tested in both synthetic and real datasets. The first column presents the rainy image, the second column shows the actual no-rain images from the synthetic dataset (no example images on the real dataset), the third column is the deraining result of the PReNet algorithm, and the fourth column is the deraining result of the R-PReNet algorithm of this paper. The two or three block images below each image enlarge the details of the images above. It can be seen that R-PReNet can reconstruct the rain-free image with clearer background structure and reduce the introduction of artifacts.

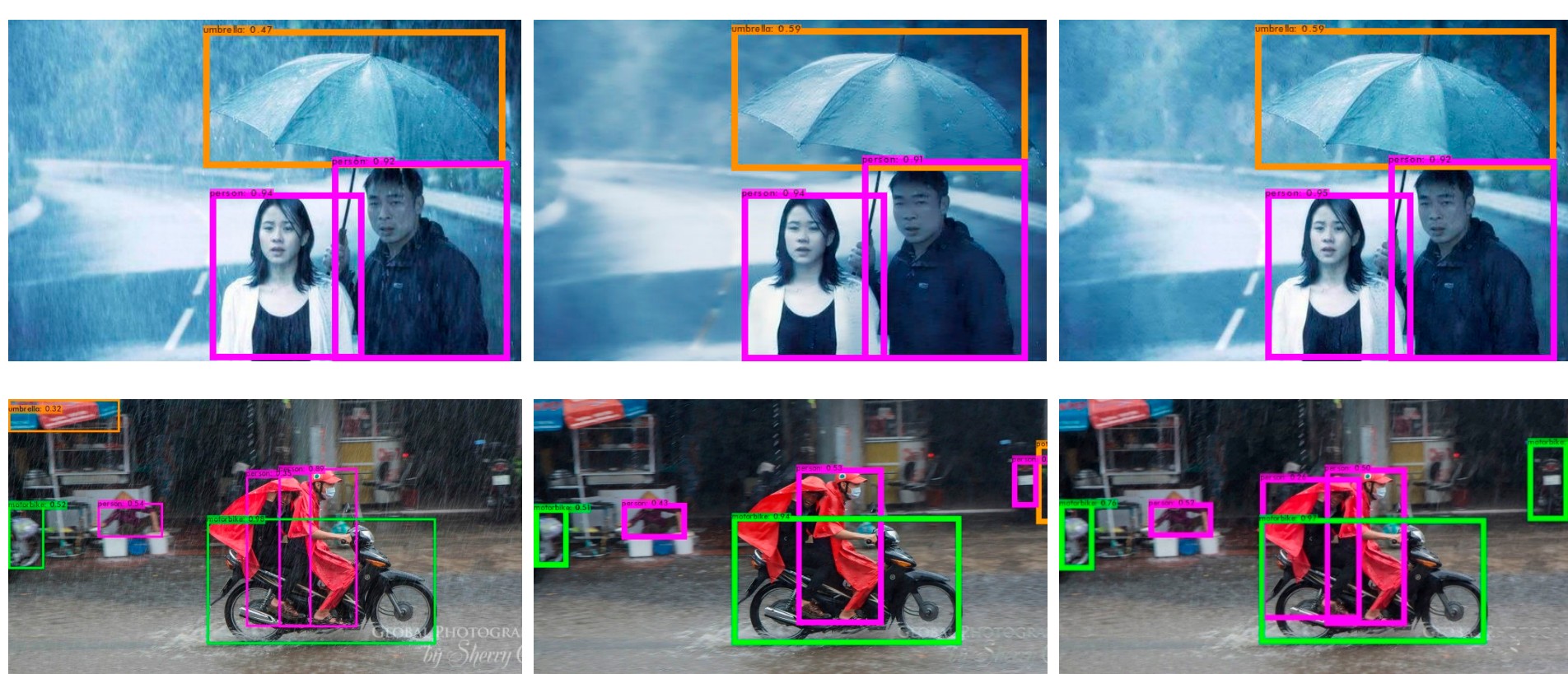

**Figure 7.** *Cont.*

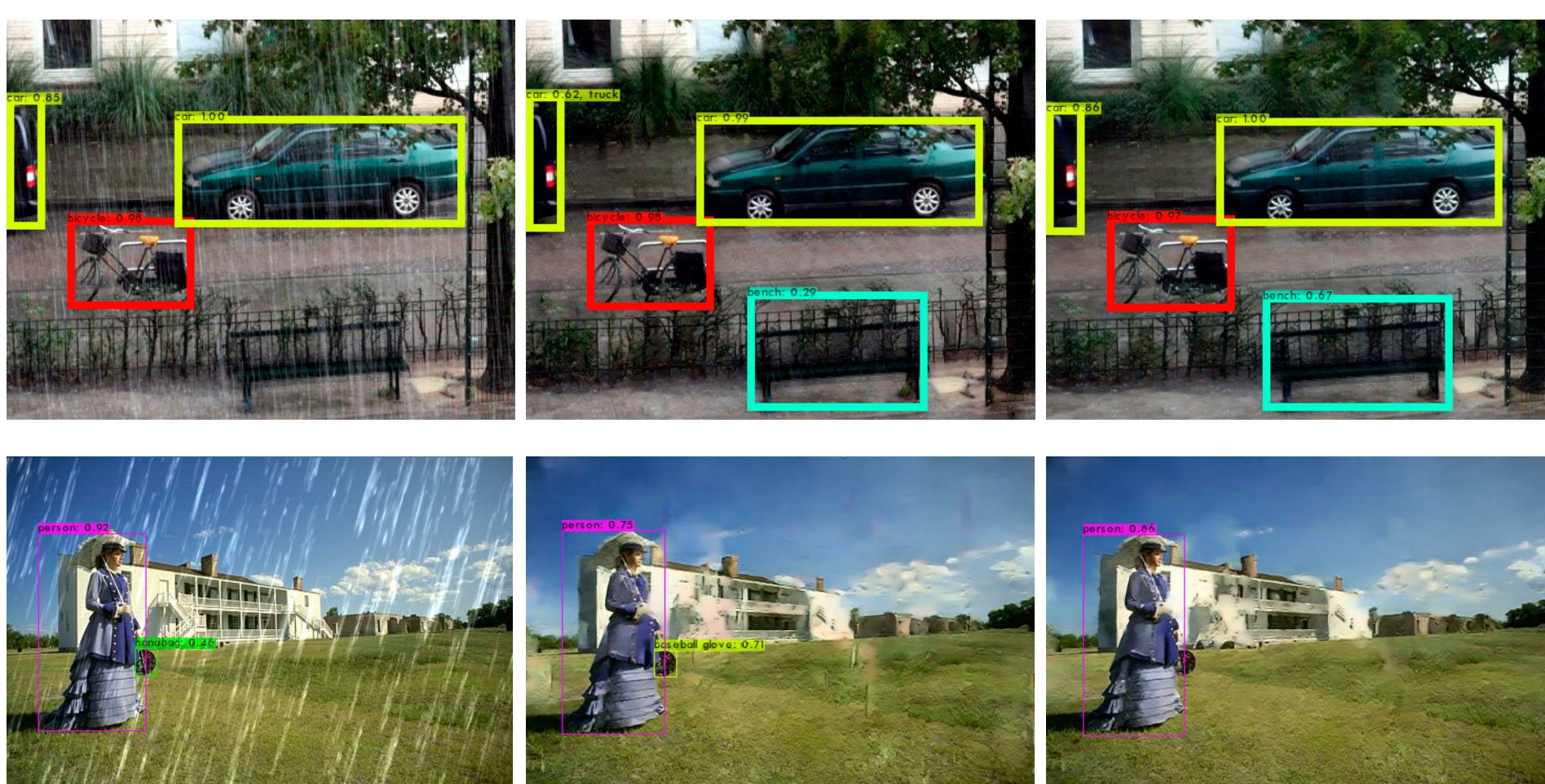

**Figure 7.** *Cont.*

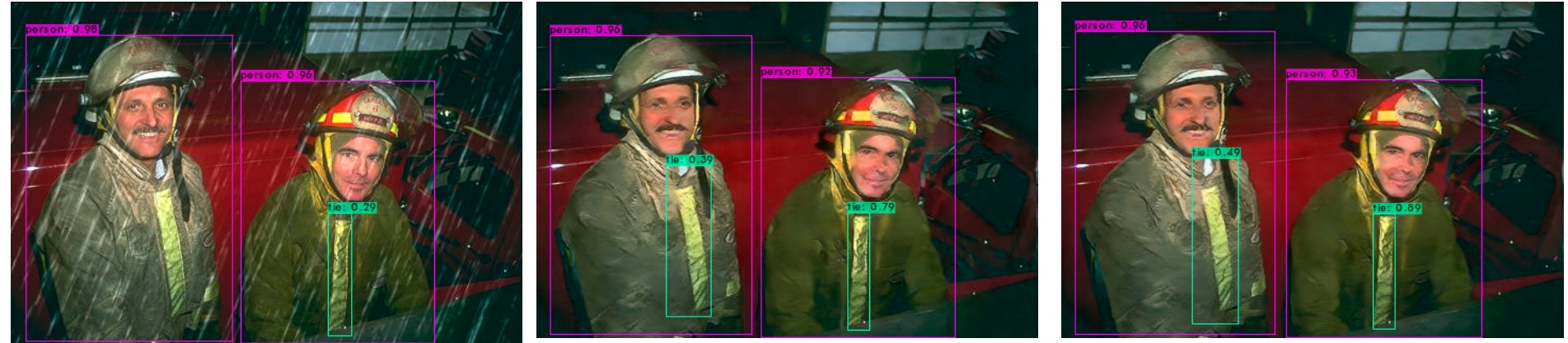

(**a**) The identification result of rain image     (**b**) The identification result of PReNet     (**c**) The identification result of R-PReNet

**Figure 7.** The identification result of the image deraining results tested in both synthetic and real datasets. The first column is the target confidence degree of target recognition after using PReNet algorithm to remove rain, and the second column is the target confidence degree of target recognition after using R-PReNet algorithm to remove rain. The recognition algorithm uses YOLOv4 algorithm for target detection and recognition, which is pre-trained on the MS COCO dataset.

*4.1. Experimental Setup*

4.1.1. Datasets

In this paper, evaluations were primarily conducted on synthetic datasets and real datasets. The synthetic image datasets included (1) Rain100L, where 200 pairs of images were used for training and 100 pairs of images were used for testing; (2) Rain100H which had 200 synthetic images used for training and 100 images used for testing; and (3) Rain14000, which was composed of training and test images with a ratio of 12,600:1400 split. The real dataset consists of (1) the Practical_by_Yang dataset with 34 images without ground-truth; and (2) 25 real rainy images from certain movie and television productions.

4.1.2. Evaluation Indicators

In these experiments, for images with ground-truths, evaluations for each method were made using two commonly adopted quantitative metrics: peak signal-to-noise ratio (PSNR) [40] and structural similarity index (SSIM) [38]. For the images without ground-truth (i.e., real dataset), visual results were provided.

*4.2. Ablation Study*

4.2.1. Effectiveness on RCP Module

The first ablation study evaluates the performance of R-PReNet with experimental results with and without the RCP module, networks with and without the RCP mode, as well as baseline algorithms JORDER [13] and RESCAN [28] were trained and tested on the Rain100L, Rain100H, and Rain14000 datasets. Table 1 shows the performance of the above algorithms on the quantitative results in PSNR and SSIM. Both quantitative and visual results show that the recurrent network with RCP module outperforms the network without RCP module and the baseline algorithm.

**Table 1.** Performance comparison of synthetic datasets on network structure with and without RCP module.

| PSNR/SSIM Methods | PReNet | R-PReNet | JORDER [12] | RESCAN [28] | DDN [14] | GMM [23] |
|---|---|---|---|---|---|---|
| Rain100H | 29.46/0.899 | 30.76/0.916 | 26.54/0.835 | 28.88/0.866 | 26.05/0.8056 | 14.50/0.4164 |
| Rain100L | 37.48/0.979 | 38.87/0.984 | 36.61/0.974 | - | 34.68/0.9671 | 28.66/0.8652 |
| Rain14000 | 32.60/0.946 | 33.03/0.963 | - | - | - | - |

4.2.2. Effectiveness on IFM Module

To investigate the effectiveness of the feature fusion module, two different network architectures were compared: (a) with the RCP module, but the RCP high-dimensional features were directly connected with the rainy image features into the network, and (b) with the RCP module and the IFM module, which used interactive fusion to combine the RCP high-dimensional features and the rainy image features together into the network. Networks with and without the FIM module, as well as baseline algorithms JORDER [13], RESCAN [28], and PReNet [7], were trained and tested on the datasets Rain100L, Rain100H, and Rain14000, respectively. Table 2 shows the quantitative results of the above algorithms in PSNR and SSIM. Both quantitative and visual results showed that the recurrent network with IFM module outperforms the network without IFM module and the baseline network.

The data in the tables are, respectively, PSNR and SSIM, where PSNR is expressed as the peak signal-to-noise ratio between the image after rain removal and the original rain image. The larger the PSNR value between the two images, the more similar it is. The general benchmark of PSNR is 30 dB, and the degradation of images below 30 dB is more obvious. The SSIM is represented here as the structural similarity between the image after the rain and the ground-truth, and the value is a floating-point number between zero and one. According to the experimental data, R-PReNet, by protecting background

information, has improved image background and details in visual effects and PSNR and SSIM quality evaluation data compared with PReNet. However, since this algorithm protects the background of the image and optimizes the rain removal effect from the details, the improvement of PSNR/SSIM will not be very large.

**Table 2.** Performance comparison of synthetic datasets with and without IFM module network structure.

| PSNR/SSIM Methods | PReNet | R-PreNet (No IFM) | R-PReNet | JORDER [12] | RESCAN [28] | DDN [14] | GMM [23] |
|---|---|---|---|---|---|---|---|
| Rain100H | 29.46/0.899 | 29.86/0.901 | 30.76/0.916 | 26.54/0.835 | 28.88/0.866 | 26.05/0.8056 | 14.50/0.4164 |
| Rain100L | 37.48/0.979 | 37.67/0.967 | 38.87/0.984 | 36.61/0.974 | - | 34.68/0.9671 | 28.66/0.8652 |
| Rain14000 | 32.60/0.946 | 32.89/0.954 | 33.03/0.963 | - | - | - | - |

## 5. Conclusions

In this paper, a progressive recursive denoising network based on background preservation is proposed. The experiments show that this algorithm can remove rain streaks and protect background information at the same time. In the preprocessing stage of rainy images, a residual channel is initially extracted from the rainy image. The extracted residual channel, devoid of rain streaks, is utilized to extract high-dimensional features. Subsequently, these extracted features are interactively fused with rainy image features and then fed into the progressive recursive network. The input for each stage of the network consists of the fused features, the reconstructed image from the previous stage, and the original rainy image. After generations of progressive recursion, the final rain-free image is produced. Comprehensive experimental evaluations show that our method outperforms the original algorithm on both synthetic and real rainy images.

**Author Contributions:** Methodology, C.J.; Software, C.J.; Investigation, F.M.; Resources, F.M.; Data curation, T.L. and Y.C.; Writing—original draft, C.J.; Writing—review & editing, F.M.; Supervision, F.M.; Funding acquisition, F.M. All authors have read and agreed to the published version of the manuscript.

**Funding:** This work was supported by the National Natural Science Foundation of China under Grant 61305040 and the Scientific Research Program Serving Local Special Projects of Shaanxi Provincial Education Department of China under Grant 23JM018.

**Data Availability Statement:** The datasets collation link used in this algorithm is updated as follows: https://pan.baidu.com/s/1o_lFQclEstiKEdCQOlaVeg?pwd=mbcz (accessed on 30 July 2023). Among them, the synthetic datasets Rain100H and Rain100L are from the public dataset, which are provided in the paper "Joint Rain Detection and Removal from a Single Image". Synthetic dataset Rain14000 is from the synthetic dataset and is provided in the paper "Removing Rain from Single Images via a Deep Detail Network"; The real dataset is provided in the paper "Joint Rain Detection and Removal from a Single Image"; The real dataset of "real_rain" is from the public images on the network, the specific link is in the "download_link" txt file.

**Conflicts of Interest:** The authors declare no conflict of interest.

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
