# Peer review of "R-PreNet: Deraining Network Based on Image Background Prior"

_applsci, doi:10.3390/app132111970_

Round 1
Reviewer 1 Report
Comments and Suggestions for Authors
The paper is well structured and the literature review is quite good.
I have two major concerns.
1) I'm not sure I could reproduce the work just based on the description in the paper. The hyperparameters for learning are not sufficiently explained. Nor does there seem to be a public repository with the implementation?
2) The evaluation is insufficient. There needs to be some objective evaluation of success using real data. Right now, the success of the method for real images is fully subjective "visual results show"... The evaluation using the artificial datasets only tells us that the method improves the results on another method when it comes to removing fake white streaks from a photograph. The only way of properly measuring the success is by using detection/recognition methods that are ultimately supposed to be improved. Such as face recognition methods or number plate recognition. Does your method improve those results on real images compared to other methods?
Right now, the results in tables 1 and 2 are not very meaningful on their own.
Maybe this is an issue in general for deraining, datasets are inherently difficult to produce. However, since in the introduction you mention the applications that are to benefit from deraining, you could use those applications to measure the actual impact of deraining.

Just some "the"s missing and some unusual phrasing, otherwise good.
Reviewer 2 Report
Comments and Suggestions for Authors
The idea of this paper is very interesting.
However, some points must be enhanced as follows:
1 - The related work must be enhanced
2- The comparative study must include others techniques(2methods are not enough)
3-The performance metrics Must be explained by mathematics formula
4- The Link Of dataset Must be shown.
This paper requires a minor revision
Reviewer 3 Report
Comments and Suggestions for Authors
Authors Have presented their work with the title “R-PreNet: Deraining Network Based on Image Background Prior”
The research work contributed by the authors of this paper is mainly reflected in the following points:
· In this Article Authors have replicates and tests the PreNet deraining network on three popular image deraining datasets and studies the results of deraining.
· This paper explores the effectiveness of residual channel prior (RCP) for background protection and proposes an image deraining network structure based on RCP.
· Numerous experiments have shown that their method outperforms the original method on commonly used rainfall datasets, restoring visually clean images and good details.
· They also proposed an RCP extraction module and an interactive fusion module (IFM) for RCP extraction and guidance, respectively, to obtain deep features of RCP and guide the network to recover more background details.
· I request the authors to avoid the words like I, WE, YOU in the paper running text and the paper should be written in third person format.
· I request the authors to add the following papers which are relevant to your work in references and cite them in the running text:
1. https://www.mdpi.com/2073-4441/15/19/3462
2. https://doi.org/10.1155/2023/3544724
3. https://www.scopus.com/inward/record.uri?eid=2-s2.0-85132786767&doi=10.1007%2f978-981-16-8739-6_3&partnerID=40&md5=8a15c3bcf73edaad60b2a50447d7326a
4. https://www.scopus.com/inward/record.uri?eid=2-s2.0-85128518037&doi=10.1007%2fs11063-021-10679-4&partnerID=40&md5=a721b35ad7b90875db9856b407d7f0b2
5. https://www.scopus.com/inward/record.uri?eid=2-s2.0-85096224355&doi=10.1007%2fs12524-020-01265-7&partnerID=40&md5=b849e51ed4724a086901f2923c055698
Comments on the Quality of English LanguageAuthors Have presented their work with the title “R-PreNet: Deraining Network Based on Image Background Prior”
· I request the authors to avoid the words like I, WE, YOU in the paper running text and the paper should be written in third person format.
